# Evaluation of Typhoid Conjugate Vaccine Effectiveness in Ghana (TyVEGHA) Using a Cluster-Randomized Controlled Phase IV Trial: Trial Design and Population Baseline Characteristics

**DOI:** 10.3390/vaccines9030281

**Published:** 2021-03-19

**Authors:** Andrea Haekyung Haselbeck, Birkneh Tilahun Tadesse, Juyeon Park, Malick M. Gibani, Ligia María Cruz Espinoza, Ariane Abreu, Craig Van Rensburg, Michael Owusu-Ansah, Sampson Twuamsi-Ankrah, Michael Owusu, Isaac Aguna, Valentina Picot, Hyonjin Jeon, Ellen Higginson, Sunju Park, Zenaida R. Mojares, Justin Im, Megan E. Carey, Farhana Khanam, Susan Tonks, Gordon Dougan, Deokryun Kim, Jonathan Sugimoto, Vittal Mogasale, Kathleen M. Neuzil, Firdausi Qadri, Yaw Adu-Sarkodie, Ellis Owusu-Dabo, John Clemens, Florian Marks

**Affiliations:** 1International Vaccine Institute, Seoul 08826, Korea; birkneh.tadesse@ivi.int (B.T.T.); juyeon.park@ivi.int (J.P.); lcruz@ivi.int (L.M.C.E.); ariane.abreu@ivi.int (A.A.); craig.rensburg@ivi.int (C.V.R.); hyonjin.jeon@ivi.int (H.J.); sunju.park@ivi.int (S.P.); zenaida.mojares@ivi.int (Z.R.M.); justin.im@ivi.int (J.I.); drkim@ivi.int (D.K.); vmogasale@ivi.int (V.M.); jclemens@icddrb.org (J.C.); 2Cambridge Institute for Therapeutic Immunology and Infectious Disease, University of Cambridge, Cambridge CB2 0SL, UK; eeh48@cam.ac.uk (E.H.); mec82@cam.ac.uk (M.E.C.); st823@cam.ac.uk (S.T.); gd312@medschl.cam.ac.uk (G.D.); 3Department of Infectious Disease, Imperial College of Medicine, London SW7 2AZ, UK; m.gibani@imperial.ac.uk; 4School of Public Health, Kwame Nkrumah University of Science and Technology, Kumasi 00000, Ghana; mikeansah@yahoo.com (M.O.-A.); sampson.ankrah@yahoo.com (S.T.-A.); michaelowusu80@gmail.com (M.O.); isaacaguna98@gmail.com (I.A.); yasax@hotmail.co.uk (Y.A.-S.); owusudabo@yahoo.com (E.O.-D.); 5Fondation Mérieux, 69002 Lyon, France; valentina.picot@fondation-merieux.org; 6International Centre for Diarrhoeal Disease Research, Dhaka 1212, Bangladesh; farhanak@icddrb.org (F.K.); fqadri@icddrb.org (F.Q.); 7Seattle Epidemiologic Research and Information Center, Cooperative Studies Program, Office of Research and Development, United States Department of Veterans Affairs, Seattle, WA 98174, USA; jons@fredhutch.org; 8Fred Hutchinson Cancer Research Center, Seattle, WA 98109, USA; 9Department of Epidemiology, University of Washington, Seattle, WA 98195, USA; 10Center for Vaccine Development and Global Health, University of Maryland School of Medicine, Baltimore, MD 21201, USA; kneuzil@som.umaryland.edu; 11Fielding School of Public Health, University of California, Los Angeles, CA 90095, USA; 12Laboratory of Microbiology and Parasitology, University of Antananarivo, Antananarivo 566, Madagascar

**Keywords:** typhoid conjugate vaccine, typhoid fever, cluster randomized trial, Ghana

## Abstract

Typhoid fever remains a significant health problem in sub-Saharan Africa, with incidence rates of >100 cases per 100,000 person-years of observation. Despite the prequalification of safe and effective typhoid conjugate vaccines (TCV), some uncertainties remain around future demand. Real-life effectiveness data, which inform public health programs on the impact of TCVs in reducing typhoid-related mortality and morbidity, from an African setting may help encourage the introduction of TCVs in high-burden settings. Here, we describe a cluster-randomized trial to investigate population-level protection of TYPBAR-TCV^®^, a Vi-polysaccharide conjugated to a tetanus-toxoid protein carrier (Vi-TT) against blood-culture-confirmed typhoid fever, and the synthesis of health economic evidence to inform policy decisions. A total of 80 geographically distinct clusters are delineated within the Agogo district of the Asante Akim region in Ghana. Clusters are randomized to the intervention arm receiving Vi-TT or a control arm receiving the meningococcal A conjugate vaccine. The primary study endpoint is the total protection of Vi-TT against blood-culture-confirmed typhoid fever. Total, direct, and indirect protection are measured as secondary outcomes. Blood-culture-based enhanced surveillance enables the estimation of incidence rates in the intervention and control clusters. Evaluation of the real-world impact of TCVs and evidence synthesis improve the uptake of prequalified/licensed safe and effective typhoid vaccines in public health programs of high burden settings. This trial is registered at the Pan African Clinical Trial Registry, accessible at Pan African Clinical Trials Registry (ID: PACTR202011804563392).

## 1. Introduction

Typhoid fever (TF), caused by the *Salmonella enterica* subspecies *enterica* serovar Typhi (*S.* Typhi), is one of the most common causes of bacteremia in several low- and middle-income countries and caused an estimated 10.9 million cases and 116,000 deaths globally in 2017 [1]. Whilst TF can affect all age groups, the incidence is higher in preschool and school-age children compared to older people [2]. TF incidence estimates suggest that south-central Asia, Southeast Asia, and southern Africa are regions with high incidence, defined as >100 cases per 100,000 person years, particularly in subregions with limited access to improved water quality and with poor access to sanitation [1,3,4,5]. Recent data from the Typhoid Surveillance in Africa Program revealed marked heterogeneity in the incidence of TF in different surveillance sites across the African continent, with some areas having rates of >100 per 100,000, including Ghana, Burkina Faso, and Madagascar [1,6,7].

Most typhoid cases are effectively treated with antibiotics, although the case fatality rate remains at about 1% [1]. Moreover, growing rates of antibiotic resistance and extensively drug-resistant *S*. Typhi strains, resistant to first-line antimicrobials chloramphenicol, ampicillin, and trimethoprim-sulfamethoxazole; fluoroquinolones; and third generation cephalosporins, in many countries reduce the number of viable treatment options and necessitate increasingly complicated and costly alternatives [8]. Improvements in water quality, sanitation infrastructure, and implementation of hygienic practices can reduce TF disease burden as has been observed in high-income countries [9]. However, the development of adequate infrastructure for improved water and sanitation requires large and long-term investments and is therefore a distant reality for impoverished populations. Instead, limited economic opportunities in rural settings have led to rapid resource-poor urbanization and unsustainable population density, significantly contributing to the risk of typhoid transmission [10]. Additionally, basic health education for handwashing and proper food handling has had measured effects in reducing TF [11], which would take time to implement in the community.

In October 2017, the WHO has recommended the use of TCV in typhoid-endemic countries from the age of 6 months onwards, administered as a single dose and programmatic administration in combination with other childhood vaccines at 9 months of age or in the second year of life [12]. A position paper has since been published in 2018 updating recommendations on typhoid vaccination, superseding a position paper from 2008 [12]. Where feasible and supported by epidemiologic data, catch-up vaccination in children and adolescents up to 15 years of age is also recommended. The position paper highlights that the introduction of TCVs should be prioritized in countries with a high burden of TF or a high prevalence of antimicrobial resistance. Several areas for further research were highlighted, particularly better understanding the duration of protection afforded by a single dose, the need for re-vaccination, and the need to identify immunological correlates of protection. TYPBAR-TCV^®^, a Vi-polysaccharide conjugated to a tetanus-toxoid protein carrier (Vi-TT), was prequalified by the WHO in January 2018 and Gavi the Vaccine Alliance has committed an $85 million funding window to support the roll out of TCV in eligible countries between 2019 and 2020.

Overall, TCVs have been shown to have a reassuringly safety profile across age groups including infants and children [13,14,15,16,17]. An individually randomized trial of Vi-TT in Nepal, conducted by the Typhoid Vaccine Acceleration Consortium (TyVAC) led by the University of Maryland in partnership with the University of Oxford and PATH, determined the vaccine efficacy to be 81.6% (95% confidence interval, between 58.8% and 91.8%), while another trial in Malawi is expected to provide estimates of efficacy within this year [18,19]. Furthermore, a cluster-randomized TyVAC trial in Dhaka, Bangladesh, will provide an estimate of the population level Vi-TT effectiveness [20].

As of January 2021, Vi-TT is licensed for use in nine countries worldwide including Tanzania, Ghana, Kenya, and Zambia. However, large-scale effectiveness data of TCVs in many African settings are still limited. Effectiveness data provide critical input to decision making by national and regional stakeholders to drive country and regional uptake in Africa, where the epidemiology of typhoid fever differs from South Asia. Nonetheless, there are no cluster-randomized trials currently ongoing on the continent and population-level protection data are largely lacking. Herein, we present the design of the first cluster-randomized trial assessing the real-world vaccine effectiveness of TCVs in preventing typhoid fever on the African continent. The data generated will be used in estimating the budgetary impact and in understanding the value of the money invested in the potential introduction of TCV in Ghana.

## 2. Materials and Methods

### 2.1. Trial Design

The Typhoid Conjugate Vaccine Effectiveness in Ghana (TyVEGHA) study is a participant- and observer-blinded, cluster-randomized phase IV trial with clusters randomized to receive Vi-TT or the control vaccine MenAfriVac (group A meningococcal polysaccharide–tetanus toxoid conjugate vaccine (MCV-A)) to evaluate the effectiveness and safety of Vi-TT in Ghanaian children and adolescents. The cluster-randomized design was selected in order to assess both the individual-level effectiveness (on vaccinees) and population-level effectiveness (on both vaccinees and non-vaccinees) including herd protection of vaccination of children and adolescents aged 9 months to <16 years living in the Asante Akim District in Ghana for a period of 3 years. The first vaccination was conducted through a cluster level mass vaccination of age eligible children (Year 1), followed by annual catch-up vaccinations (Year 2 and 3) for age-eligible children and adolescents at the trial sites. Enhanced passive surveillance was conducted in parallel in the trial catchment area to identify blood-culture-confirmed TF cases.

This trial is registered at the Pan African Clinical Trial Registry, accessible at www.pactr.org (ID: PACTR202011804563392).

### 2.2. Cluster Randomization and Method of Blinding

The TyVEGHA study population was recruited from a geographically defined catchment area in the Asante Akim District of Ghana (see Figure 1, TyVEGHA catchment area). The baseline map of the area was developed in partnership with the Statistical Service office in Agogo, Ghana. Additional data from a baseline census were used to inform details on the map and to confirm the catchment area boundaries for the trial. The census population data were merged with the Geographic Information System mapping (GIS,) data for cluster formation (QGIS.org, 2021. QGIS Geographic Information System. QGIS Association.). The study area was divided into 80 clusters. Each cluster consisted of a total population of 400–600 inhabitants at average, with an estimated proportion of 40% in the eligible age group for TyVEGHA. The entire age-eligible cluster population was eligible for vaccination and analized under primary analysis. In addition, a buffer region formed at the border of each cluster, was drawn to insulate adjacent clusters, minimizing cross-cluster transmission [21] if necessary. The age-eligible individuals residing in the inner region served as the population under exploratory analysis.

GIS mapping tools were used for identification of all geographical characteristics and demarcation of clusters [22]. Randomization lists allocated each cluster to receive either Vi-TT or the control vaccine MCV-A in equal numbers (40 clusters in each arm), based on population size, using stratified block randomization. Stratification was by clusters, the distance to the closest healthcare facility (above the median versus at or below the median), and the percentage of children and adolescents aged 9 months to <16 years age (above the median versus at or below the median).

Once participants will be enrolled in the study, they will be vaccinated with either the trial or the control vaccine according to the residential cluster in which they live. Census updates were conducted annually to identify birth and mortality rates as well as in- and out-migration in the defined clusters. The census was designed to capture (1) socioeconomic information of the households, including profession, household income, and level of education and (2) healthcare-seeking behavior and to access to water, sanitation, and hygiene (WaSH) infrastructure. Household verbal consent for the census was provided by the head or a key informant of the household.

### 2.3. Study Participants

A total of 23,000 eligible children and adolescents will be vaccinated—11,500 each in the Vi-TT and MCV-A arms. A subset for immunogenicity assessment will include 600 children and adolescents aged 9 months to <16 years (1:1 ratio by age strata; <5 years and ≥5 years, individually randomized 2:1 ratio by intervention and comparator), and a subset for safety monitoring will include 3000 children and adolescents aged 9 months to <16 years (individually-randomized, 1:1 ratio by intervention and comparator) (see Figure 2, trial flowchart).

To be eligible to participate in the study, the individual will be healthy, aged 9 months to <16 years (i.e., ≤15 years and 364 days) at the time of vaccination, voluntarily provide consent via their parents or legally authorized representative (LAR), and live within the study cluster at the time of vaccination.

Individuals will not be enrolled if they, or a parent or LAR, report a history of allergic reactions to any vaccine component, ongoing acute and/or chronic illness, or any coagulopathies (e.g., bleeding tendencies); if there are any compelling medical or social reasons in the judgment of a physician; if they are pregnant (including self- or parent-/LAR-reported or positive urine testing) or are lactating; or if they received a typhoid vaccination (proven by presentation of a vaccine card or parent-/LAR-reported) in the last 5 years prior to presentation.

Participants will be temporarily excluded if, at the time point of vaccination, the parent/LAR reported fever (elevated tympanic (≥38 °C) or axillary temperature (≥37.5 °C)) in the child within 24 h and/or the use of antipyretics within 4 h prior to vaccination or if they received any other vaccination during the previous 4 weeks (confirmed by vaccine card or parent-/LAR-reporting). Females ≥11 years of age with self- or parent-/LAR-reported irregular menstruation or who did not know their last menstruation date will be reassessed and consented at or after their next menstrual period. A reassessment at least 48 h after the initial screening of resolving the condition of temporary exclusion criteria will be conducted to ensure these temporary exclusion criteria no longer existed.

### 2.4. Vaccination and Immunogenicity Assessment

The trial staff identified potential participants living in the 80 clusters of Asante Akim North, Ghana. Trial staff, including community health volunteers, will systematically approach each household in the area to identify households with children and adolescents that will be age-eligible at the time of vaccination. Once a household with a potential participant will be identified, the parent/LAR of the child will be given basic information about the trial and invited to attend the nearest study vaccination clinic.

After arrival at the study clinic, participants and their parents/LAR will be provided with an explanation of study objectives and invited to participate following written informed consent. Assent will be sought for participants ≥11 years of age. Once consent and assent (as appropriate) will be obtained, trial staff will screen volunteers according to the inclusion/exclusion criteria and conduct a medical examination, including measuring baseline temperature to assess for temporary exclusion criteria. Demographic information (including age and address) and participant/parent/LAR contact details will be updated during this visit. Participants or their parent/LAR meeting at least one temporary exclusion criteria after the medical assessment will be informed of the reason and asked to return in ≥48 h after resolving the condition of temporary exclusion criteria for repeat assessment and re-consenting.

Consented participants will be randomized by cluster to receive either Vi-TT or MCV-A. The randomization list by cluster will be computer-generated prior to the start of the vaccination campaign and will be provided to the staff assigned for randomization to reduce discriminatory practices or unfair assignment of vaccines. Based on the cluster randomization, the appropriate vaccine will be administered by a trained member of the study team. The site of vaccination will be recorded. If the participant’s parent/LAR consent to participation in the immunogenicity substudy, a blood sample will be collected prior to vaccination by a trained staff member. The participant will be considered enrolled in the study at the point in that any medical procedure took place (i.e., blood sample collected or vaccination administered).

A study card will be provided during the annual update census, which contains the name of the study, the participant ID, and contact details for the study team will be given to all participants/parents/LAR for use if they have any concerns or if their child is admitted to hospital at any time during the study. The study card will also contain instructions for attending the study health care facilities or sentinel sites if the participant develops a fever at any time over the follow-up period of three years.

A subset of 600 participants will be selected on a 2:1 ratio (Vi-TT vs. MCV-A) to have blood samples collected at baseline (Day (D) 0) and at D28, D365, D730 and D1095 post-vaccination to assess vaccine immunogenicity. Safety assessments will include an active safety follow-up contact on day 7 to monitor adverse events following immunization in participants in the safety substudy (*n* = 3000, 1:1 assignment to intervention and control group) using study diary cards, and 3 years of passive hospital-level febrile and severe adverse event (SAE) surveillance. An independent Safety Monitoring Committee (SMC) was established to assess safety results and to correspond with the Data Safety Monitoring Board (DSMB). Randomization lists for these subsets will be generated prior to the vaccination campaign by an independent statistician who is not part of the study. The selection of these participants were age-stratified (<5 years vs ≥5 years of age, with an allocation ratio of 1:1) (Figure 1). Participants who will be enrolled in the immunogenicity substudy will receive additional face-to-face follow-up visits at the Agogo healthcare facility. At these additional visits, blood samples will be collected along with information relating to fever episodes or SAEs since the previous contact.

### 2.5. Trial Outcomes

The primary outcome of interest is the total protection of single-dose Vi-TT vaccination, measured as the incidence of blood-culture-confirmed typhoid fever in vaccinated children and adolescents, comparing intervention with the control clusters. Secondary outcomes include the occurrence of adverse events reported from participants receiving Vi-TT compared with MCV-A through active safety follow-up in a subset (safety subgroup) within the first 7 days post-vaccination and passive safety follow-up of serious adverse events within 6 months of vaccination determined by self-reporting at follow-up contact. The incidence of blood-culture-confirmed symptomatic TF, clinical TF (cases with fever persisting for >72 h), and severe TF, defined as blood-culture-confirmed TF with any one systemic complication of TF detected at presentation, will be assessed in all residents (overall protection), comparing intervention clusters with control clusters as well as in vaccinated individuals (total protection) and in unvaccinated individuals (indirect protection) [23]. The short-term (D28) and long-term (D365, D730, and D1095) seroconversion rates and antibody response against *S.* Typhi (anti-Vi IgM and IgG) and invasive non-typhoidal Salmonella after Vi-TT administration will be determined at defined timepoints.

Exploratory outcome measures include the assessment of incidence of clinically diagnosed intestinal perforations that are associated with *S*. Typhi, incidence of acute abdominal surgery and surgical procedure on intestinal perforations, and all-cause hospitalization rates in all residents and in vaccinees of the intervention cluster compared with the control clusters. The analysis of mortality rates assessed through verbal autopsy in the study area differentiates between all-cause mortality, mortality with fever, and mortality with fever and abdominal pain between the intervention clusters compared with the control vaccine clusters. In addition, the prevalence of antimicrobial-resistant *S.* Typhi pre- and post-vaccination will be compared between the intervention and control clusters, and the corresponding antimicrobial prescription patterns were observed in participating health care facilities that are part of the existing surveillance system.

### 2.6. Sample Size/Statistical Methods

The sample size was calculated by assuming an expected cumulative incidence rate of typhoid fever in control clusters over 3 years of 36 cases per 10,000 person-years in a catchment population of 21,213 i.e., 45,961.5 person-years of risk. Based on a preliminary analysis of survey data from the target population, we assumed that 50% of febrile illness cases seek healthcare at Agogo Presbyterian Hospital [24]. A total of 80 clusters were formed in 15 villages considering the natural settlement of the population and administrative borders. The current average number of children and adolescents aged 9 months to less than 16 years per village was 287 based on the population census in 2019 (see Table 1). We estimated the vaccine coverage at 65% and a total effectiveness of 80% with a coefficient of variation of 0.5 for a one-sided test of significance at an alpha of 0.05.

Following Hayes and Moulton, a total sample size of 13,200 person-years, which was obtained by sampling 80 clusters (40 in each arm) with an average cluster size ranging from 145 to 180, achieved >80% power to detect a total effectiveness of 80% between the intervention event rate 0.0007 and the control event rate 0.0036 [25]. The between-cluster coefficient of variation in the control and treatment groups was assumed as 0.500. A two-sided t-test of the event-rate difference was used with a significance level of 5%. The total sample sizes ranging from 11,600 to 14,400 adjusted for 15% drop-out rate were within target eligible population size considering 65% vaccine coverage.

For the immunogenicity substudy, a total of 600 children and adolescents will be enrolled, stratified by age (<5 years and ≥5 years). Each age stratum was randomized 2:1 to receive the intervention (*n* = 200) or control vaccine (*n* = 100) with an assumption of a 90% reference seroconversion rate, −10% of non-inferiority margin, and 25% drop-out rate.

For the safety monitoring substudy, a total of 3000 children and adolescents with a 1:1 ratio of each arm will be enrolled according to the “rule of three” to detect at least one event with 1500 participants in each arm. The upper limit of 95% CI of the incidence rate yielded 0.2% in each arm [26].

The primary outcome measure is blood-culture-positive TF detected during passive surveillance (see Section 3) at the study sites in vaccinees of the inner clusters demarcated geographically at baseline. To evaluate the heterogeneity of Vi-TT vaccine protection among the different subgroups, we will evaluate the interaction terms between the vaccination and subgroup variables in the models. *p*-values for the analyses will be calculated as two-tailed. Subgroup analyses will include different age stratifications (<5 years and ≥5 years; <2 years and ≥2 years) and gender (male *versus* female). The subgroup analysis was not powered to evaluate the primary outcome, and any observed patterns were interpreted cautiously, owing to the large study population and increased chance of type I error.

For safety assessment in the subset for active follow-up, solicited and unsolicited adverse events (AEs) at day 7 post-vaccination will be assessed. The frequency and proportion of participants who experience solicited AE and unsolicited AE post-vaccination will be presented with the 95% CI. Additionally, the number of participants and number of occurrences of AEs by vaccine arm for each age strata as well as overall ages were assessed. The occurrence of any SAE within 6 months follow-up post-dose will be also summarized and listed.

An immunogenicity analysis will be presented as the geometric mean titre (GMT) with 95% CI at baseline (D 0), D 28, D 365 D 730, and D 1095, and as a proportion of four-fold increase from baseline (seroconversion rate) with 95% CI at D 28, D 365, D 730, and D 1095.

Data defined by the study protocol will be captured in a clinical study database using REDCap [27]. All clinical and laboratory data were entered from paper forms into the electronic data capture (EDC) system. Data will be entered using tablets, laptops, or computers into a web-based version of the electronic case report forms (eCRFs). Edit checks programmed within the eCRFs will validate the data at the time of data entry. Performance qualification of the eCRFs will be performed, in particular on the edit checks. Data entry and subsequent data handling will be performed by trained study staff under the supervision of the site investigators.

## 3. Outcome Measurement

### Severe Typhoid Fever in Africa (SETA) Plus Surveillance System

Surveillance for enteric fever and clinically suspected perforations was integrated within the Severe Typhoid Fever in Africa (SETA) Program [23]. This established TF surveillance system is expanded to include a tertiary health care facility and a community-based referral system for all residents in the intervention and control clusters as part of the new SETA Plus surveillance program, which is subsequent to SETA and enrolled patients since March 2020 (see Figure 3). Surveillance activities will continue for three years. Febrile patients living in defined catchment areas including the TyVEGHA study area presenting with elevated tympanic (≥38°C) or reporting fever for ≥3 consecutive days within the week before seeking care at the study healthcare facility (HCF) were asked to participate and data on the fever episode were collected. Patients with clinically suspected TF or with surgically confirmed gastrointestinal perforations were also included. Blood culture, malaria testing, diagnostic microbiology, and antimicrobial susceptibility testing were performed as part of routine clinical care. A blood-culture-confirmed case was defined as *S*. Typhi isolated from a positive blood culture. Patients enrolled in SETA Plus received appropriate initial treatment, with follow-up of blood-culture-positive cases including informing patients of the results and adjustment of treatment based on laboratory findings in accordance with hospital guidelines.

## 4. Cost Effectiveness Analysis

Health economics studies are planned to transform evidence generated from TyVEGHA to support decision making during the introduction of TCVs in Ghana. As a first step, we will project the TCV delivery cost in Ghana based on the country’s immunization practices using a Microsoft Excel-based tool (Microsoft Corporation. Microsoft Excel 2018, available from: https://office.microsoft.com/excel), which was used to estimate the costs of TCV vaccination in India [28]. The tool deploys a micro-costing approach to identify all activities needed for vaccination, including the respective prices and quantities, and to project the financial (direct expenditures) and economic costs (financial costs as well as the monetized value of the additional donated or existing items). National program and financial managers will be consulted during the delivery cost projection excecise. Secondly, we will conduct a budget impact analysis using data generated from the delivery costing tool to estimate the amount of money needed to implement various vaccination strategies in Ghana, including effcts on the overall immunization budget. Thirdly, we will analyze the value for money invested in various TCV vaccination strategies based on the cost-effectiveness analysis using broader principles recommended by the World Health Organization (WHO) [29]. The economic analysis will use typhoid disease burden and cost of illness data generated through the surveillance program [23,30], and delivery costs and vaccine effectiveness generated through TyVEGHA. The cost-effectiveness analysis will compare various scenarios of vaccination strategies (for example, age *versus* geographically targeted) against the current situation of no vaccination. In-country immunization managers will be consulted to gather information on vaccination strategies included in the analysis.

## 5. Baseline Characteristics of the Study Population

The baseline census was conducted in 2019 to enumerate all households within the geographically defined catchment area. The census field team visited each structure and updated all identified geographic features including GPS coordinates. The field team also identified all structures that house any health care services as well as nonresidential structures. Demographic information was collected on all household members, such as total size of household, age and sex of all household members, and family relationship.

A total of 13,226 households were visited, and their GPS coordinates were collected. The age distribution showed a generally young population, with around 40% (22,973/55,881) between 9 months up to 16 years of age (see Table 1). The gender ratio was 46.5% males to 53.5% females. A total of 626 pregnancies were enumerated, indicating a birth rate ofof 11 per 1000 population.

The survey on water, sanitation, and hygiene (WaSH) factors showed that the main drinking water sources were at the community level, including unprotected systems such as boreholes and wells (29%) or surface water (3%) (see Table 2). Few households have access to a potable water supply on their premises (6.3%) or use packaged water (32.9%). Treating water prior to drinking, either by boiling or filtering, was practiced by 11% of households. Around 60% of households reported that they always washed their hands using soap and water after defecation.

Regarding access to sanitation facilities, adults from 59.4% of the surveyed households reported using latrines. The main sanitation option reported from households with children (*n* = 7327) were latrines (29.1%) and open defecation (19%).

## 6. Discussion

This protocol described a cluster-randomized study designed to measure population-based vaccine effectiveness of the Vi-TT vaccine in Ghana, embedded in a blood-culture-based typhoid fever surveillance system. The TyVEGHA trial constitutes one arm of the Typhoid Conjugate Vaccine Introduction in Africa (THECA) program (www.thecaproject.net). The first population census conducted in 2019 defined the baseline cohort profile, including demographic characteristics and key baseline WaSH indicators. We observed that access to safe water and hygiene practices was poor while access to sanitation facilities was reported in two thirds of adults and a one third of children. Over 40% of the catchment population was found to be younger than 16 years, those who are considered eligible for vaccination in the trial.

Our study complements the efforts on effectiveness evaluation of WHO-prequalified TCVs and those under clinical development to support introduction in public health programs. To date, two TCVs have been WHO prequalified for use in children–Vi-TT (Typbar-TCV^TM^, Bharat Biotech Hyderabad India Ltd., Hyderabad, India) in 2018 and Vi-CRM_197_ (Vi-Polysaccharide conjugated CRM197 protein, BioMed E) in 2020 [31,32]. Additionally, two programs evaluating Vi-DT (Vi-polysaccharide-conjugated diphtheria toxoid) conjugate approaches (SK Bioscience, Seongnam, Korea and PT BioFarma, Bandung, Indonesia) are in late-stage clinical development and are expected to receive WHO prequalification by 2022. Vi-DT was shown to elicit strong immune responses and has a very strong safety profile [16,17,33]. There is ongoing discussion within the THECA consortium to use one of these newly prequalified TCV candidates in a mass vaccination campaign followed by a prospective cohort study to estimate real-life effectiveness in other African countries.

Data generated from TyVEGHA describing the real-world effectiveness of one of the WHO prequalified vaccines will be of particular importance to policy makers in high-burden countries and will provide them with a clearer view of the full public health value of TCVs deployed at this scale as they develop typhoid control and elimination strategies. Furthermore, it paves the way for more impact evaluation studies of different TCVs in high-burden settings. Health economic studies including vaccine delivery cost estimation, budget impact analysis, and cost-effectiveness analyses provide an evidence summary that is directly applicable to Ghana and readily usable for making decisions on TCV introductions. The results of TyVEGHA may be transferable to similar settings on the African continent and may provide evidence to promote further uptake of TCVs by health authorities in typhoid-endemic countries.

## Figures and Tables

**Figure 1 vaccines-09-00281-f001:**
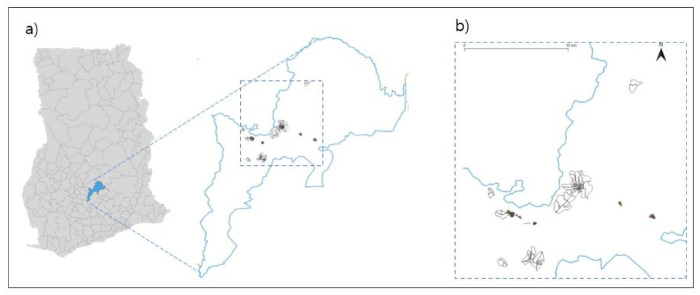
TyVEGHA catchment area. Figure (**a**) depicts the TyVEGHA catchment area located in Asante Akim North in Ghana; Figure (**b**) illustrates the demarcation of the TyVEGHA enumeration areas.

**Figure 2 vaccines-09-00281-f002:**
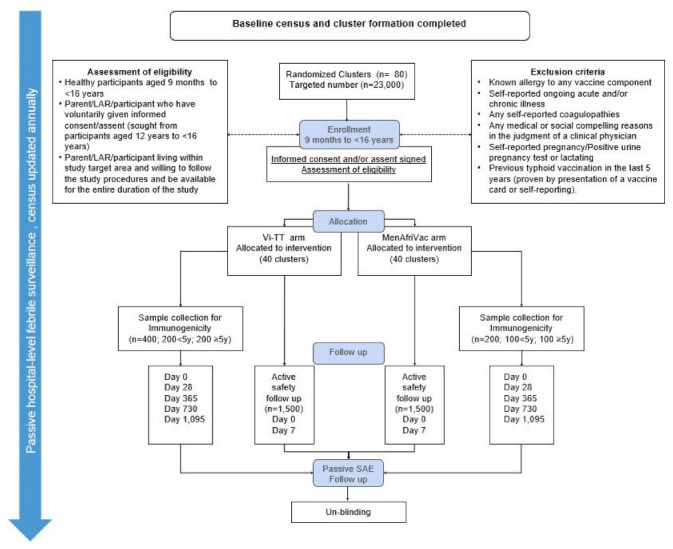
Trial flowchart.

**Figure 3 vaccines-09-00281-f003:**
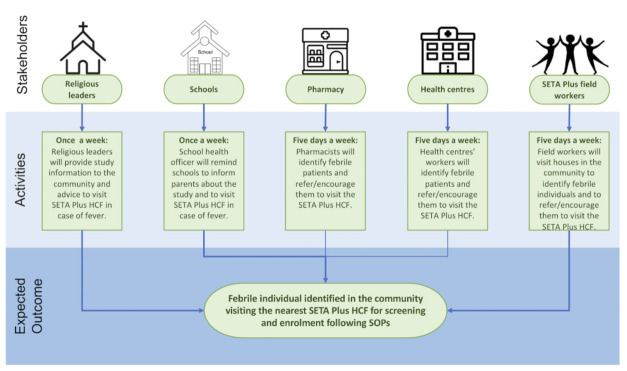
Community-based referral system during SETA Plus surveillance period. Notes. HCF, health care facility; SOP, Standard Operating Procedures.

**Table 1 vaccines-09-00281-t001:** Total number of households and household members.

		Total (%)	Male (%)	Female (%)
**Number of Households**		13,266		
Number of Household Members		55,881 (100%)	26,002 (46.53%)	29,879 (53.47%)
Age Group	<2 Year	2804 (5.02%)	1412 (50.36%)	1392 (49.64%)
	2 to <5 Years	4488 (8.03%)	2251 (50.16%)	2237 (49.84%)
	5 to 15 Years	15,225 (27.25%)	7741 (50.84%)	7484 (49.16%)
	≥15 Years	33,364 (59.71%)	14,598 (43.75%)	18,766 (56.25%)
Age Group (Eligibility)	<9 Months	1051 (1.88%)	546 (51.95%)	505 (48.05%)
	9 Months to <16 Years	22,973 (41.11%)	11,628 (50.62%)	11,345 (49.38%)
	≥16 Years	31,857 (57.01%)	13,828 (43.41%)	18,029 (56.59%)
Number of Pregnant Women		626 (2.10%)		

**Table 2 vaccines-09-00281-t002:** Water, sanitation, and hygiene (WaSH) behaviour and safe water access.

Variables	N = 13,226 *n* (%)
**Main Source of Drinking Water**	
Own source	834 (6.29)
Tap	403 (3.04)
Well	266 (2.01)
Borehole	165 (1.24)
Communal Source	7655 (57.70)
Tap	3808 (28.70)
Well	359 (2.71)
Borehole	3488 (26.29)
Bottled/Sachet Water	4377 (32.99)
River/Stream/Pond/Dugout	400 (3.02)
**Drinking Water Treatment**	
Boiled	67 (0.51)
Filtered	1275 (9.61)
Boiled and Filtered	153 (1.15)
Neither Boiled nor Filtered	11,746 (88.54)
Do not Know	25 (0.19)
**Access to Water/Soap After Defecation**	
Always	7036 (53.04)
Never	3357 (25.31)
Sometimes	2861 (21.57)
Do not know	12 (0.09)
**Use of Water/Soap After Defecation**	
Always	7990 (60.23)
Never	524 (3.95)
Sometimes	4714 (35.53)
Do not know	38 (0.29)
**Toilet Facility Used for Adults**	
Flush Toilet Used Alone by Household	1029 (7.76)
Indiscriminate/Free Range	899 (6.78)
Flush Toilet Shared with Other Households	1260 (9.5)
Latrine	7885 (59.44)
Other ^†^	2193 (16.53)
**Toilet Facility Used for Under 5 Children ^‡^**	
Flush Toilet Used Alone by Household	293 (4.00)
Indiscriminate/Free Range	1415 (19.31)
Flush Toilet Shared with Other Households	259 (3.53)
Latrine/Kid Latrine	2135 (29.14)
Other ^§^	1837 (25.07)

^†^ Other: public toilet, public KVIP (Kumasi Ventilated Improved Pit); ^‡^ N = 7327, total number of households which have children under 5; ^§^ other: diaper (Pampers), chamber pot.

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
