# Peer review of "Evaluation of Typhoid Conjugate Vaccine Effectiveness in Ghana (TyVEGHA) Using a Cluster-Randomized Controlled Phase IV Trial: Trial Design and Population Baseline Characteristics"

_vaccines, 2021, doi:10.3390/vaccines9030281_

Round 1
Reviewer 1 Report
The paper entitled “Evaluation of Typhoid Conjugate Vaccine Effectiveness in 2 Ghana (TyVEGHA) using a cluster-randomized controlled 3 Phase IV trial: Trial design and population baseline characteristics” by Haselbeck and collaborators displayed the features of a clinical trial to evaluate the safety and efficacy of a typhoid conjugate vaccine in Ghana. In addition, another goal of the study is to provide strong data for policy makers in order to solve this important public health issue. The topic is relevant. Typhoid fever, caused by Salmonella enterica subspecies enterica serovar Typhi, is a common disease in poor countries, and it affects several millions of people around the world. The disease affects all ages but is particularly higher in children than adults. It’s a public health problem that can be solved through the water quality improvement, sanitation and hygiene practices. However, these solutions require large investments. In another hand, vaccines can be an important and effective tool to use in this battle.
The Overall, the manuscript is well-written, the tools/methods are robust, and the future result will impact the field. The manuscript is a description of a future clinical trial and this is the reason why I did not include here any comments about the findings. Nonetheless, the possibility to make this information available to peers around the world put the future study and the authors in a really unique position. They can receive several feedbacks that will refine the study and improve the power of conclusion. All that before the beginning of the clinical trial. This feature combined with all the methods described are the strengths of the present study. I suggested only minor modifications to the authors as inclusion of extra figures or tables, and to check the quality of some figures (1 and 2). In my opinion this will provide a better understanding of the study.
- Please, go over the text and make sure that all the abbreviations are described in the first appearance. I noticed that in the line 32 (Vi-TT) and line 248 (iNTS).
- Check the quality of figure 1 and 2. The letters looks small and a little blurred.
- Can the criteria describe in lines 165-184 presented as a table? Maybe it could be included as part of the main figures or as supplementary data.
- Can the author include a figure displaying the region map and the clusters delimitation? I think it will very helpful.
Reviewer 2 Report
I was invited to review the paper entitled "Evaluation of Typhoid Conjugate Vaccine Effectiveness in Ghana (TyVEGHA) using a cluster-randomized controlled Phase IV trial: Trial design and population baseline characteristics". This paper aimed to report baseline charactestics of a cluster randomized trial evaluating vaccine effectiveness of TCV in Ghana. It was an outstanding paper that improve the knowledge on this field and it is of great interests for the reader.
The paper is clear and easy to read and background was deeply presented.
Methodology was strong and deeply described.
I have only two minor questions for Authors:
- Among secondary outcomes, Why did Authors not considered also typhoid pneumonia and multiorgan disease? I know that they are rare complications but they can develop in worst outcomes.
- In my opinion, in discussion sections Authors should also compare this vaccine with oral live attenuated vaccine.
